# Presence and Persistence of Putative Lytic and Temperate Bacteriophages in Vaginal Metagenomes from South African Adolescents

**DOI:** 10.3390/v13122341

**Published:** 2021-11-23

**Authors:** Anna-Ursula Happel, Christina Balle, Brandon S. Maust, Iyaloo N. Konstantinus, Katherine Gill, Linda-Gail Bekker, Rémy Froissart, Jo-Ann Passmore, Ulas Karaoz, Arvind Varsani, Heather Jaspan

**Affiliations:** 1Department of Pathology, Institute of Infectious Diseases and Molecular Medicine, University of Cape Town, Anzio Road, Cape Town 7925, South Africa; anna.happel@uct.ac.za (A.-U.H.); frk.balle@gmail.com (C.B.); iyaloombodo@gmail.com (I.N.K.); jo-ann.passmore@uct.ac.za (J.-A.P.); 2Seattle Children’s Research Institute, 307 Westlake Ave. N, Seattle, WA 98109, USA; Brandon.Maust@seattlechildrens.org; 3Department of Pediatrics, University of Washington School of Medicine, 1959 NE Pacific St., Seattle, WA 98195, USA; 4Namibia Institute of Pathology, Hosea Kutako, Windhoek 10005, Namibia; 5Desmond Tutu HIV Centre, University of Cape Town, Anzio Road, Cape Town 7925, South Africa; katherine.Gill@hiv-research.org.za (K.G.); Linda-Gail.Bekker@hiv-research.org.za (L.-G.B.); 6NRF-DST CAPRISA Centre of Excellence in HIV Prevention, 719 Umbilo Road, Congella, Durban 4013, South Africa; 7CNRS, IRD, Université Montpellier, UMR 5290, MIVEGEC, 34394 Montpellier, France; Remy.FROISSART@cnrs.fr; 8National Health Laboratory Service, Anzio Road, Cape Town 7925, South Africa; 9Earth and Environmental Science, Lawrence Berkeley National Laboratories, 1 Cyclotron Rd., Berkeley, CA 94720, USA; ukaraoz@lbl.gov; 10The Biodesign Center of Fundamental and Applied Microbiomics, Center for Evolution and Medicine, School of Life Sciences, Arizona State University, 1001 S. McAllister Ave., Tempe, AZ 85281, USA; 11Structural Biology Research Unit, Department of Integrative Biomedical Sciences, Institute of Infectious Diseases and Molecular Medicine, University of Cape Town, Anzio Road, Cape Town 7925, South Africa; 12Department of Global Health, University of Washington School of Public Health, 1510 San Juan Road NE, Seattle, WA 98195, USA

**Keywords:** bacteriophages, prophages, lytic, vaginal microbiota, stability, CRISPR, Sub-Saharan Africa

## Abstract

The interaction between gut bacterial and viral microbiota is thought to be important in human health. While fluctuations in female genital tract (FGT) bacterial microbiota similarly determine sexual health, little is known about the presence, persistence, and function of vaginal bacteriophages. We conducted shotgun metagenome sequencing of cervicovaginal samples from South African adolescents collected longitudinally, who received no antibiotics. We annotated viral reads and circular bacteriophages, identified CRISPR loci and putative prophages, and assessed their diversity, persistence, and associations with bacterial microbiota composition. *Siphoviridae* was the most prevalent bacteriophage family, followed by *Myoviridae*, *Podoviridae*, *Herelleviridae*, and *Inoviridae*. Full-length siphoviruses targeting bacterial vaginosis (BV)-associated bacteria were identified, suggesting their presence in vivo. CRISPR loci and prophage-like elements were common, and genomic analysis suggested higher diversity among *Gardnerella* than *Lactobacillus* prophages. We found that some prophages were highly persistent within participants, and identical prophages were present in cervicovaginal secretions of multiple participants, suggesting that prophages, and thus bacterial strains, are shared between adolescents. The number of CRISPR loci and prophages were associated with vaginal microbiota stability and absence of BV. Our analysis suggests that (pro)phages are common in the FGT and vaginal bacteria and (pro)phages may interact.

## 1. Introduction

Bacteriophages are one of the most abundant and diverse biological entities on earth, and they play an important role in shaping the structure of bacterial communities and in contributing to the evolution of bacterial genomes [1,2,3]. Diversity and dynamics of bacteriophage community composition have been associated with various adverse health outcomes in humans, including the ability of opportunistic bacterial pathogens to establish in the gut [4,5], pathogenesis of inflammatory bowel disease [6,7,8], immunodeficiency in human immunodeficiency virus (HIV) disease [9], and severity of respiratory tract infections [10]. Both direct and indirect ecological interactions have been proposed to contribute to these outcomes. Bacteriophages can directly lyse their hosts, releasing progeny bacteriophages (then referred to as virulent or bacteriophages with a lytic cycle), or can incorporate their viral genomes into the host cell genome (then referred to as temperate bacteriophages with a lysogenic cycle, and their integrated genome referred to as prophages) [11]. Local environmental factors may influence the frequencies in which bacteriophages transition from a lysogenic to a lytic life cycle [12,13,14,15]. The switch from lysogeny to a lytic lifecycle of bacteriophages contributes to shaping bacterial communities and favours the dissemination of antibiotic resistance genes and other mobile genetic elements [16,17,18,19]. However, cryptic prophages modulate genetic diversity and functionality of bacterial communities [3,19] and provide fitness benefits to their bacterial host [20,21,22,23,24,25], as well as protection against secondary infection by closely related bacteriophages that belong to the same immunity group [26,27,28]. All these mechanisms may shape bacterial community structure over time.

In the human gut, the collective genome content of (pro)phages, referred to as the “phageome”, constitutes a substantial proportion of the genetic diversity [29,30,31,32,33,34]. Virulent bacteriophages have been shown to modulate the enteric bacterial microbiota composition [9,35,36], and expansion of bacteriophages has been linked to immune cell expansion and increased enteric inflammation [7,36,37,38]. Virulent bacteriophages have also been shown to have high nucleotide substitution rates [29], indicating rapid evolution, while temperate bacteriophages showed relatively lower substitution rates, consistent with replication by high-fidelity bacterial DNA polymerases in the integrated prophage state [29,39,40,41]. This might explain the high persistence of prophages compared to virulent bacteriophages [29,30,42]. Thus, presence of prophages in the human microbiota could contribute to the stability of the phageome and bacteriome over time [43]. Whether the prevalence of prophages and female genital tract (FGT) microbial stability are related remains to be determined. Prophage sequences have been identified in genomes of various vaginal *Lactobacillus* spp. using in vitro and whole genome sequencing approaches [13,44,45,46,47]. Recent metagenomic surveys of the women from high-income countries have revealed the presence of known *Lactobacillus* bacteriophages in the vagina [48,49], suggesting that they can be induced and actively transmitted in the FGT. No in-depth analysis on (pro)phages in vaginal metagenomes from African women has been conducted, in whom vaginal bacterial diversity is relatively high. Furthermore, little is known about the frequency of occurrence, stability, and genetic variability of prophages within the FGT.

In the current study, we performed a metagenomic sequence-derived survey of (pro)phages in vaginal samples of South African adolescent girls and young women and investigated their persistence, presence in multiple participants, diversity, antibiotic resistance gene carriage, and likely functionality. Furthermore, we explored associations with bacterial microbiota composition and stability.

## 2. Materials and Methods

### 2.1. Study Cohort

The Division of AIDS and the University of Cape Town (UCT) Health Science Human Research Ethics Committee (HREC) approved the UChoose Trial (ClinicalTrials.gov reference number NCT02404038), which was conducted in full compliance with South African Good Clinical Practice (SA-GCP), ICH76 GCP, and ICMJE guidelines. The UCT HREC also approved this sub-study (HREC 801/2014). Eligibility criteria and study design are described in detail elsewhere [50]. Briefly, 130 nonpregnant HIV-seronegative females aged 15–19 years were enrolled from Cape Town, South Africa, and followed up longitudinally every 4 months over 8 months, in a randomised study of injectable hormonal contraception (norethisterone enantate, NET-EN), combined contraceptive intravaginal ring (CCVR, NuvaRing^®^; MSD Pty Ltd., Johannesburg, South Africa), or combined oral contraceptive pills (COCP, Triphasil^®^ or Nordette^®^).

A rapid HIV and a pregnancy test were conducted at every study visit. If either test was positive, the participant was referred for counselling and clinical management, and no further samples were collected. Medical history, sexual behaviour, menstrual cycle, contraceptive use, intravaginal practices, and antibiotic use were assessed using interviewer-assisted questionnaires. As described previously [51], vulvo-vaginal swabs were collected for sexually transmitted infection (STI) testing (*Chlamydia trachomatis*, *Neisseria gonorrhoeae*, *Trichomonas vaginalis*, *Mycoplasma genitalium*) by multiplex PCR, Nugent scoring, and pH measurement, and a lateral wall swab for 16S rRNA gene and shotgun metagenomic sequencing. Upon arrival at the laboratory, vaginal swabs were stored at −80 °C until testing. No samples were collected during menstruation.

### 2.2. Sample Selection for This Sub-Study

Longitudinal samples from 13 participants were selected for this sub-study. Criteria for selection included not having taken any antibiotics or medication that might have influenced the microbiota throughout the study period nor 40 days prior to enrolment, nor practicing any vaginal insertion practices (including washing with water and/or soap, insertion of traditional or other medicines, cloth, tampons, and douching), nor having been diagnosed with an STI during the study and having attended all study visits. Participants with bacterial vaginosis (BV) by Nugent Scoring were not excluded. The participant IDs (PIDs) used here corresponded to those used in a prior study that evaluated the bacterial microbiota by 16S rRNA gene sequencing [51].

### 2.3. DNA Sample Preparation and Whole Community Metagenomic Sequencing

Genomic DNA extraction methods have been reported previously [51]. DNA was shipped to the University of Washington Northwest Genomics Center (NWGC, USA) for metagenomic sequencing. Starting with a minimum of 135 ng of DNA, all DNA from individual samples were sheared in a 96-well format using a Covaris LE220 focused ultrasonicator targeting 350 to 380 bp fragments. One sample did not reach this minimal DNA quantity and was thus excluded from sequencing. The resulting sheared DNA was cleaned with Agencourt AMPure XP beads to remove sample impurities prior to library construction. A two-sided AMPure cleanup was performed to further restrict the fragment sizes to the desired range. End-repair, A-tailing, and ligation were performed as directed by KAPA Hyper Prep Kit without amplification (KR0961 v1.14) protocols. Two AMPure clean-ups were performed after ligation to remove excess adapter dimers from the library. The resulting libraries were amplified with 9 cycles of PCR. A final AMPure clean-up was performed. Concentrations were quantified via an Invitrogen HS DNA Qubit and fragment sizes were analysed with an Agilent Fragment Analyzer SS. All library construction steps were automated on the Perkin Elmer Janus platform. Prior to sequencing, final library concentration was determined by triplicate qPCR using the KAPA Library Quantification Kit (KK4824), and molecular weight distributions were verified using the Agilent Bioanalyzer. Samples were sequenced using the Illumina NovaSeq 6000 Reagent Kit (v1.0). Cluster generation was performed on a cBot modified for use with the NovaSeq flow cells.

### 2.4. Sequence Processing and Annotation

As described previously, base calls were generated instantaneously on the NovaSeq 6000 (RTA 3.1.5), whereafter demultiplexed, unaligned BAM files were generated by Picard ExtractIlluminaBarcodes and IlluminaBasecallsToSam. Finally, BAM files were aligned to a human reference (hg19hs37d5) using BWA-MEM (Burrows-Wheeler Aligner; v0.7.10) [52]. Subsequently, all non-human read data were subject to the following steps: (1) “duplicate removal” (i.e., the removal of reads with duplicate start positions; Picard MarkDuplicates; v1.111), (2) indel realignment (GATK IndelRealigner; v3.2), resulting in improved base placement and lower false variant calls, and (3) recalibration of base qualities (GATK BaseRecalibrator; v3.2). Remaining reads were assembled using metaSPAdes [53]. Taxonomic assignment of de novo assembles contigs was performed using GOTTCHA2 [54]. To evaluate the presence of viral reads, contigs determined to correspond to circular molecules based on terminal redundancy were analysed using BLASTx against the NCBI viral Refseq database. The Integrated Microbial Genomes Viral database (IMG/VR) and analysis server version 2 was used to call open reading frames (ORFs) and annotate sequences [55]. The bacteriophage sequence data presented in this study were submitted to NCBI under BioProject PRJNA767784.

### 2.5. In Silico Identification of Virulent Bacteriophages in Vaginal Metagenomes

Circular bacteriophages were annotated using RAST [56], and hosts were predicted with PHISDectector [57]. We searched sequences against the NCBI *Caudovirales* database to identify similarities with previously described bacteriophages and used vConTACT v.2.0 [58] for taxonomic assignment.

### 2.6. In Silico Identification and Characterisation of Prophages in Vaginal Metagenomes

Prophage sequences from all metagenomic contigs that were longer than 1000 nts were predicted with VirSorter (v1.0.3, CyVerse implementation with default options using the VirSorter Refseq database) [59]. Results were filtered to include only “confident” hits (defined as whole regions that were enriched for viral-like genes or non-Caudovirales genes and had at least one hallmark viral gene present) and “likely” hits (regions that either were enriched for viral-like or non-Caudovirales genes or had a viral hallmark gene present, associated with at least one of the following other virus-predicting metrics: enrichment in short or uncharacterised genes, depletions in PFAM affiliated genes, or strand switch) in downstream analyses. Sequences of prophage-like elements were aligned to the metagenomic contigs from which they originated, and putative hosts were identified by searching flanking sequences against the NCBI bacterial RefSeq database using BLASTx [60]. All metagenomic reads were mapped to the sequences of prophage-like elements, and their percentage coverage was recorded. To evaluate the diversity of prophages within a certain predicted host, multiple alignment of all sequences from same host was performed, and phylogeny was established using FastTree 2.1. [61] to identify clusters of closely related prophage sequences. Interactive Tree of Life (iTOL) Version 5 [62] was used to display the phylogenetic trees. Viral proteomic trees for classification of *G. vaginalis* putative prophages based on genome-wide sequence similarities with reference viruses contained in the Virus–Host database [63] computed by tBLASTx were generated using ViPTree 1.9.1. [64]. To assign putative functions to the prophages, coding sequences predicted by IMG/VR [55] were used to generate a sequence similarity network (SSN) using the Enzyme Similarity Tool (EFI-EST) [65] using default options with an E-value of 1 × 10^−5^. The network was visualized using Cytoscape Version 3.8.1 [66], and protein clusters of bacteriophage hallmark genes [67] and potential antibiotic resistance genes were colour-coded. Translated predicted prophage coding sequences were aligned against the Comprehensive Antibiotic Resistance Database (CARD) [68] and NCBI’s Bacterial Antimicrobial Resistance Reference Gene Database using Geneious Prime^®^ 2020.1.2 (Biomatters Ltd., Auckland, New Zealand).

### 2.7. Identification of CRISPR Repeats/Spacers, Predicted Cas Proteins, and CRISPR-Cas Systems

The CRISPR tool in Geneious Prime^®^ 2020.1.2 (default settings, limiting searches to arrays with at least 3 spacers) and the CRISPRCasFinder [69] were used to identify CRISPR arrays. Only CRISPR arrays that were identified by both tools were included in subsequent analyses. To gain perspective on the potential bacteriophage (and plasmid) sequences in our dataset, we aligned the predicted, extracted CRISPR spacer sequences against the IMG/VR [55] and the CRISPRCasFinder [69] CRISPR spacer databases, and against the metagenome assembled genomes (MAGs) from our own dataset using Geneious Prime^®^ 2020.1.2 (Biomatters Ltd., New Zealand).

### 2.8. Data Analysis

All downstream analysis and generation of figures were conducted in RStudio, unless otherwise specified. Study cohort characteristics were described using means, medians, standard deviations, and proportions, as appropriate. Differences in study population characteristics were tested using Pearson’s chi-squared test or Fisher’s exact test (when the expected value was <5) for count measurements and unpaired Student’s t-test for differences in mean (parametric data) and unpaired Mann–Whitney U test for differences in medians (nonparametric data), with post hoc testing for continuous measurements. Community clusters were identified using Manhattan distance. All *p* values < 0.05 were considered statistically significant.

## 3. Results

### 3.1. Cohort Characteristics

Of the 130 adolescents enrolled in the UChoose trial, 13 had not taken any antibiotics or medication that might have influenced their microbiota throughout the study period nor in the 40 days prior to enrolment, did not report any vaginal insertion practices, were not diagnosed with an STI, and had attended all study visits and were thus included in this sub-study. An overview of participants’ demographics, medical, and reproductive history (including age, body mass index (BMI), Nugent-BV, and bacterial community state types (CSTs) based on 16S rRNA gene amplicon sequencing [51]) is provided in Table 1. Demographic and biological characteristics remained comparable throughout the 32-week study period; only the use of injectable hormonal contraceptives at the time of sampling decreased, while use of COC and CCVR increased due to the nature of the parent study.

### 3.2. Vaginal Metagenomes of South African Adolescents

Shotgun metagenomic sequencing was performed on 38 samples from 13 participants, collected at three time points 16 weeks apart (baseline, week 16, and week 32). A total of 3.3 × 10^9^ reads were obtained, with an average of 13 Gb per sample. Almost 87.5% of the reads were mapped to the human genome and were removed for downstream analysis. Read-based analysis showed that the longitudinal vaginal microbial metagenomes of most participants were dominated by bacteria, with only two participants (UC084 and UC096) having a higher relative abundance of viruses than bacteria at two time points, specifically papillomaviruses (family *Papillomaviridae*) (Figure 1). Other viruses detected included those in the viral families of eukaryote-infecting *Herpesviridae* (human gammaherpesvirus 4, also known as Epstein–Barr virus) and prokaryote-infecting viruses in the families *Siphoviridae*, *Podoviridae*, and *Myoviridae*, including previously described inducible *Lactobacillus* prophages KC5a (host: *L. crispatus*) and ϕjlb1 and ϕahd (host: *L. gasseri*).

The similarity of the metagenome communities at baseline was assessed using the relative abundance of reads that were classified to non-human taxa, and the metagenomes were clustered hierarchically using Manhattan distance. Three distinct community clusters at baseline were apparent: C1 contained metagenomes dominated by a diverse range of BV-associated bacteria, C2 those dominated by *Lactobacillus iners*, and C3 those dominated by a diverse range of microbes, including *Lactobacillus* and BV-associated bacterial spp. and viruses (Figure 2A). At baseline, the metagenomes of participants in community clusters C1 and C3 tended to have a higher within-sample diversity (as measured by Shannon Index) compared with those in C2 (Figure 2B). About half of the participants who were in community cluster C2 at baseline (3/7; 42.9%) remained within this cluster over the 32-week study period, while transitions from C1 or C3 to another community cluster were observed more frequently (Figure 2C).

### 3.3. Identification of Prokaryote-Infecting Viruses in the Metagenomes of South African Adolescents

To elucidate the presence of prokaryote-infecting viruses in the vaginal metagenomes, we annotated contigs longer than 1000 nts using the NCBI viral Refseq database. Sequences from bacteriophages in five different families were identified in the vaginal samples. Members of the *Siphoviridae* family were the most prevalent, being present in all but one sample. Other identified bacteriophage families included *Myoviridae*, *Podoviridae*, *Herelleviridae*, and *Inoviridae* (Figure 3A, Table 2). In all adolescents, a bacteriophage from at least one family was detected at any given time point, and the number of bacteriophage families that were present within a given sample varied from one to four.

*Lactobacillus* bacteriophages have been previously described [70], but there is a paucity of data regarding lytic bacteriophages infecting BV-associated bacteria. Figure 3B shows the circular (indicating full-length) siphoviruses identified in vaginal metagenomes from adolescents with Nugent-BV. A bacteriophage identified in participant UC015 of 39,829 nts in length encodes a range of bacteriophage hallmark gene products, such as integrase, holin, lysin, terminase, tail fibre, capsid, and portal proteins, and its predicted host is *G. vaginalis*. Identified hosts of other bacteriophages included *A. vaginae*, *Megasphaera* genomospecies type 1, and *Coriobacteriales* bacterium. None of these bacteriophages were identified at more than one time point in cervicovaginal secretions, suggesting low persistence of these lytic bacteriophages in the FGT. To further characterise these bacteriophages, we aligned their sequences against those available in the NCBI *Caudovirales* database, but none of the identified bacteriophages showed any similarity, indicating that these vaginal bacteriophages from BV-associated bacteria have not previously been described. We also used vConTACT 2 [58] to assign taxonomy to these putative bacteriophages but were unable to do so with confidence, further supporting the novelty of the identified bacteriophages. The best hit for all these bacteriophages was uncultured phage WW-nAnB, which was previously identified by deep sequencing of viral particles from raw sewage [71]. Therefore, in addition to previously described *Lactobacillus* bacteriophages, our data indicate that virulent bacteriophages targeting BV-associated bacteria are present in the FGT and may thus contribute to changes in vaginal bacterial microbiota composition or host response. Isolation of these bacteriophages from cervicovaginal secretions and further characterisation is needed to be able to assess morphology and assign taxonomy.

### 3.4. Presence of CRISPR Loci

To further evaluate the extent to which vaginal bacteria are exposed to bacteriophages (and/or plasmids), we screened for the presence of CRISPR arrays. We identified a median of 39 CRISPR arrays (IQR 34–50) per metagenome, with a median of 350 spacers (IQR 281–418) and a total of 13,926 spacers in the complete dataset. After adjusting for read counts, participants with Nugent-BV at their final visit (n = 4/12) had fewer CRISPR arrays per million reads (median 2.9 (IQR 1.3–4.0) vs. 4.8 (IQR 4.1–7.2); *p* = 0.0182) and fewer spacers per million reads (median 27 (IQR 14–34) vs. 41 (IQR 32–59); *p* = 0.0727) than metagenomes from participants without Nugent-BV (n = 8/12). These data suggest higher previous exposure to bacteriophages and/or plasmids in adolescents without BV, possibly making them less susceptible to bacteriophage-induced perturbations of their vaginal microbiota.

We aligned the extracted spacer sequences against the Joint Genome Institute (JGI) and CRISPR/CAS9 spacer databases to identify their targets, specifically bacteriophages that might have previously infected the bacteria. Using this approach, only 166 of the 13,926 spacers could be identified (1.2%), and most of those were sequences of previously identified CRISPR loci (Appendix A). Next, we aligned the spacer sequences against the NCBI *Caudovirales* RefSeq database and were able to identify the targets of four spacers: one targeted a non-coding sequence of *Streptococcus* bacteriophage Javan11 and three targeted the tape measure protein or a hypothetical protein of *Lactobacillus* bacteriophage Lv-1. Twenty-four spacers matched circular viral sequences identified in our own dataset; however, the taxonomy of these viral sequences could not be further identified. The high number of CRISPR loci in the cervicovaginal metagenomes of South African adolescents indicates frequent exposure to bacteriophages and/or plasmids, yet the limited number of available sequenced bacteriophages in databases hampers our ability to identify their targets.

### 3.5. Identification of Putative Prophages in the Metagenomes of South African Adolescents and Associations with Vaginal Microbiota Stability

We next evaluated the presence of prophage-like elements within the metagenomes, as the occurrence of prophages can impact bacterial fitness [72,73] and may thus affect stability of the vaginal microbiota. We identified a total of 519 distinct prophage-like elements within the dataset using VirSorter, which were integrated into the genomes of *L. crispatus* (n = 165), *L. iners* (n = 117), *G. vaginalis* (n = 58), *L. jensenii* (n = 31), *L. gasseri* (n = 5), *Prevotella* spp. (n = 13), and a range of other bacterial species (Figure 4). VirSorter identified an average of 12 putative prophages per sample (range 0–30), and the host bacteria that were identified reflected the bacterial community composition of a given sample, with *L. crispatus* prophages primarily being present in adolescents without Nugent-BV and *G. vaginalis* prophages being present in adolescents with Nugent-BV (Figure 4). No clear separation was observed by type of hormonal contraceptive, but we acknowledge that our small sample size might not have allowed the detection of any differences.

To evaluate the persistence of prophages over time, we mapped the metagenome reads to all identified putative prophages (Figure 5A). A prophage was defined as being persistent when “identical” putative prophages (limited to ≥99% required nucleotide identity) were detected at ≥2 time points. We detected persistent prophages in 12/13 participants (92.3%), of which 6/12 participants had identical prophages present at all three time points. Of note, participants with a stable vaginal microbiota over time (defined as remaining either Nugent-BV positive or Nugent-BV negative throughout the 32-week study period; n = 7) tended to have a higher number of identical prophages present at several time points (median 22, IQR 11–27) compared with those who experienced a change in Nugent-BV status (n = 3; median 7, IQR 5–12; *p* = 0.1049). None of the six participants who had identical prophages present at all three time points experienced a change in Nugent-BV status throughout the study period. This suggests a high persistence of prophages over time in the FGT, unless vaginal microbiota composition changed between visits. These results further suggest that the presence of prophages within the genomes of vaginal bacteria may contribute to the stability of the vaginal microbiota, as it has been previously shown that occurrence of prophages within bacterial genomes can provide fitness benefits [20,21,22,23,24,25] and protection against secondary bacteriophage infection [26,27,28], both of which might allow host bacterial persistence and avoid disruption of the vaginal microbiota.

To further evaluate the role of prophages in BV, we compared the number of prophages in adolescents who did not have Nugent-BV (n = 8) with those who had Nugent-BV (n = 4) at the final study visit (Figure 5B). The number of prophages per million microbial reads sequenced tended to be higher in adolescents without Nugent-BV (median 16, IQR 11–19) compared with those who had Nugent-BV (median 9, IQR 4–14; *p* = 0.0727). In a paired comparison (Figure 5C), adolescents who remained Nugent-BV negative from week 16 to week 32 (n = 7) did not experience a change in prophage number per 1 million reads (median 16, IQR 15–19 vs. median 14, IQR 12–19), while those who changed Nugent-BV status (n = 2) tended to have less prophages per 1 million reads when they had BV (median 5, IQR 4–5) compared with when they did not (median 11, IQR 7–15). Whether this increase in prophages reflects the increased bacterial *Lactobacillus* strain diversity in the vaginal microbiota of adolescents who remain free of BV, or whether *Lactobacillus* spp. generally have more prophages present in their genomes compared with BV-associated bacteria needs to be further investigated.

Finally, we also observed that prophages (with ≥99% nucleotide identity) were shared between participants, with 96 putative prophages identified by VirSorter (96/519; 18.5%) having been present in multiple participants (Figure 6). Putative prophages that were identified in cervicovaginal secretions of at least two participants included 37 of the 165 (22.4%) in the dataset-identified prophages within *L. crispatus* genomes, 10/117 (8.5%) *L. iners*, 13/58 (22.4%) *G. vaginalis*, 13/45 (28.9%) prophages with unknown bacterial host, 10/31 (32.3%) *L. jensenii*, 5/15 (35.7%) *Aerococcus christensenii*, 3/13 (23.1%) *Prevotella*, 1/8 (12.5%) *Megasphaera*, 1/5 (12.5%) *L. gasseri*, 1/3 (33.3%) *Olsenella*, 1/1 (100%) BVAB1, and 1/1 (100%) *Collinsella aerofaciens* prophages (Figure 6A). Almost half of the putative prophages that were identified in more than 1 participant were only shared between 2 participants (40/95, 42.1%), while two putative *L. iners* prophages were present in all but 2 of the 13 participants. The median number of participants sharing a prophage differed significantly by a given bacterial host (Figure 6B, ANOVA *p* = 0.0314) and ranged from 5 (IQR 2–7) for *L. crispatus* to 2 for *L. jensenii* (IQR 2–3) prophages. Collectively, these results suggest that prophages, and thus likely bacterial strains, are shared between individuals in the same geographic location.

### 3.6. Genomic Characterisation of Putative Prophages

The size of prophage-like elements from *L. crispatus* (n = 165; median 12,625 nts, IQR 9122–18,371 nts) was comparable with *L. jensenii* putative prophages (n = 31; median 13,596 nts, IQR 10,032–24,301 nts) but significantly smaller than those from *G. vaginalis* (n = 58; median 22,786 nts, IQR 13,272–44,260 nts; *p* < 0.0001) and *L. iners* (n = 117; median 53,980 nts, IQR 1783–93,056 nts). As *L. iners* [74] and *G. vaginalis* [75] genomes tend to be smaller than *L. crispatus* genomes, this indicates that a larger proportion of *L. iners* and *G. vaginalis* genomes are made up of prophage-like elements.

To evaluate the diversity of prophages within each identified bacterial host, we conducted pairwise comparisons of prophage sequences and generated unrooted phylogenetic trees with minimum-evolution subtree-pruning-regrafting and maximum-likelihood nearest-neighbour interchanges [61], which depicted five major clusters for *L. crispatus* prophages, three distinct clusters for *L. jensenii* prophages, and two distinct clusters for *L. iners* prophages (Figure 7A–C). Several participants contributed to each *L. crispatus* and *L. jensenii* prophage cluster, further confirming that similar putative prophages are shared between participants. More in-depth analysis of the *L. crispatus* prophage clusters showed that if a participant had putative *L. crispatus* prophages present, she commonly carried prophages of several clusters, which might indicate that a woman carries several *L. crispatus* bacterial strains at a given time point (Figure 7A).

In contrast to the *Lactobacillus* prophages, the 64 *G. vaginalis* prophages did not form any clusters (Figure 8A), suggesting that *G. vaginalis* prophages are more diverse than *Lactobacillus* prophages, which might be reflective of their respective host strain diversity. In an attempt to further classify these putative *G. vaginalis* prophages based on genome-wide sequence similarities with viruses contained in the Virus–Host database, we generated a viral proteomic tree using ViPTree [64] (Figure 8B). Neither virus family nor host group could be assigned, suggesting that these putative *G. vaginalis* prophages are likely to be novel and not yet contained in any database. It is interesting that these prophage sequences that were integrated into *G. vaginalis* strains did not only cluster among Actinobacteria, the bacterial class of *Gardnerella*, but also among clusters of Firmicutes and Gammaproteobacteria.

### 3.7. Evaluation of Likely Functionality of Putative Prophages

To estimate whether any of the putative prophages identified by VirSorter were likely to be capable of moving from a lysogenic to a lytic cycle, we aligned the prophage sequences against the NCBI *Caudovirales* database. We were unable to identify any full-length alignments. The best hit was a *Streptococcus* prophage (identified in participant UC062) that had 70% similarity to *Streptococcus* bacteriophage Javan112 but that only covered one-third of its genome.

For a more unbiased approach, we analysed the proteins encoded by the prophages using a sequence similarity network. As expected, prophage proteins clustered by function (Figure 9). Annotation of the proteins identified a range of previously defined hallmark bacteriophage proteins [67], including terminase, capsid, integrase, tail, baseplate, head–tail connector, holin, XRE-family, HTH domain, portal protein, helicase, DNA primase/helicase (DnaB), DNA polymerase B & A, DNA gyrase B, DNA topoisomerase IV, DNA ligase, HNH endonuclease, ribonucleotide reductase, and others. Bacteriophage anti-repressor proteins that govern the switch from a lysogenic to lytic cycle were also highly abundant, as were proteins involved in SOS response, such as RecT/RecF family and response regulator proteins. While this does not ultimately confirm the capability of the putative prophages to escape their host and reinfect a different bacterial strain, the presence of viral hallmark genes suggests that at least some of the identified prophages are likely to be functional.

### 3.8. Evaluation of Antibiotic Resistance Gene Presence in Putative Prophages

The spread of antibiotic resistance via horizontal gene transfer is of interest given the high number of prophages in our dataset and the clinical implications of resistant bacterial strains. While the similarity network analysis indicated the presence of potential antibiotic resistance proteins (e.g., ABC transporters, efflux pumps, beta-lactamase, and bacitracin resistance proteins), an alignment of all translated prophage genes against the CARD and NCBI antimicrobial resistance databases did not identify any antibiotic resistance proteins, indicating that none of the 519 putative prophages identified in this study carried known antimicrobial resistance genes associated with confirmed clinically relevant antibiotic resistance, albeit we only included participants here if they had not received antibiotics 40 days prior to or during the study.

## 4. Discussion

While the role of bacteriophages at other mucosal body sites has been described extensively [9,21,30,35,36], their role in the FGT and in sexual and reproductive health remains to be determined. Here, we report results of an exploratory study that identified prevalence and persistence of (pro)phages in the FGT of 13 South African adolescents and associations with vaginal bacterial community composition. Using shotgun metagenomic sequencing, we identified bacteriophages belonging to the *Siphoviridae*, *Myoviridae*, *Podoviridae*, *Herelleviridae*, and *Inoviridae* families. These bacteriophage families are also amongst the most prevalent bacteriophages families present at other mucosal body sites, including the gut, oral cavity, and lung [76]. Bacteriophages in the *Siphoviridae*, *Myoviridae*, and *Podoviridae* families have previously been identified in urinary samples [77,78] and recently also in vaginal samples [48,49]. We also identified members of *Herelleviridae*, which is not surprising, as members of the *Twortvirinae* family infect *Lactobacillus* spp. [79]. Similarly, bacteriophages belonging to the *Inoviridae* family infect Gram-negative bacteria, including the genus *Escherichia* [80], which is also commonly found in the FGT. The species and families of some viral reads remain unidentified since our study, just as any other, is limited by the available bacteriophage sequences deposited in public databases. This emphasizes the need to make data publicly available to help improve and develop viral databases.

Prophages have previously been identified in BV-associated bacteria [81], yet to our knowledge, we are first to describe full-length bacteriophages that presumably infect BV-associated bacteria. Our untargeted sequencing-based study suggests that virulent bacteriophages targeting BV-associated bacteria are indeed present in the FGT, and determination of near-complete or complete viral genomes such as ours enables a broader understanding of viral evolution and allows classification of these viruses in a rather dynamic taxonomic framework [19,82]. It was somewhat surprising that we were not able to assign taxonomy using vCONTACT 2. However, considering that none of these putative siphoviruses aligned with those in the NCBI *Caudovirales* database, this might suggest that the databases are biased towards siphoviruses infecting other genera, and more emphasis should be placed on sequencing FGT bacteriophages, given the importance of vaginal microbiomes in sexual and reproductive health.

In agreement with in vitro studies [44,45,47,81], the prophages identified in this study were widely distributed among vaginal bacteria—both in species associated with optimal vaginal environments and in those associated with BV. We were able to describe the diversity of these prophages in detail. *G. vaginalis* prophages were highly diverse, which might represent the high diversity among bacterial *G. vaginalis* strains [81,83,84]. Compared with *L. iners* prophages, *L. crispatus* and *L. jensenii* prophages were less diverse, which again likely reflects the relatively lower diversity of these bacterial species. How prophages contribute to phenotypical and ecological differences of vaginal bacterial strains, and which ecological factors influence diversity within the vaginal microbiota, remains to be determined.

We found that prophages integrated in predominant FGT bacterial taxa were highly persistent in the cervicovaginal secretions of a given participant over time. This is in agreement with observations from the gut, where prophages were found to persist longer compared with their lytic counterparts [29,30,42]. We also observed that putative prophages seem to be nearly identical between participants. While it has previously been described that gut bacterial strains can be shared between cohabiting family members [85,86] and that vaginal *Lactobacillus* spp. strains are shared between female sex partners [87], to our knowledge this is the first indication that FGT bacterial strains may also be shared between adolescents of the same community, even in the absence of cohabitation. To investigate the sharing of FGT bacterial strains between adolescents in more detail, the bacterial shotgun metagenomic sequencing data of this project can be leveraged, which allows for the identification of microbial strains with single nucleotide variants and tracking of strains.

Although we described in the same cohort that the use of the combined contraceptive vaginal NuvaRing^®^ resulted in higher microbial diversity compared with the use of injectable contraceptive and the oral pill [51], we were unable to see differences in prophage numbers by contraceptive type in this sub-study. Most likely, this is due to the small sample size included here, and larger studies should address the question of whether type of contraceptive use influences (pro)phage occurrence in the FGT.

An expansion in abundance of members of the *Caudovirales* family has been associated with microbial dysbiosis in the gut [6], and it was hypothesised more than 20 years ago that BV is caused by an expansion of *Lactobacillus*-targeting bacteriophages [88], yet robust clinical evidence to support this hypothesis is lacking. While we cannot claim any causality with our study by its design, we observed a higher number of CRISPR loci in metagenomes of adolescents without BV than those with BV, indicating higher previous infection with bacteriophages or plasmids. This observation suggests that previous exposure to bacteriophages results in increased bacterial community stability and lower susceptibility to subsequent bacteriophage infections. We also observed a higher number of prophages in the metagenomes from adolescents without BV, providing preliminary evidence that prophages might contribute to microbiota stability in the FGT. It remains to be determined how (pro)phages influence bacterial community composition (or vice versa), and whether demographic or behavioural characteristics (e.g., a woman’s smoking status) influence this interaction. Our results are also based on a limited sample size, and the contribution of prophages to genital health needs to be tested in longitudinal studies with more frequent sampling before, during the onset, and during episodes of BV.

## 5. Conclusions

We identified novel bacteriophages of BV-associated bacteria in the FGT. Our preliminary data further suggest that prophages, and thus bacterial strains, are shared between non-cohabiting adolescents. The number of CRISPR loci and pro(phages) in the FGT might be associated with the absence of BV in South African adolescents. These associations, and ultimately causality, need to be verified in future studies. Isolation of virulent bacteriophages from cervicovaginal secretions and genomic and functional characterisation thereof would allow in-depth investigation of the role of bacteriophages in women’s health.

## Figures and Tables

**Figure 1 viruses-13-02341-f001:**
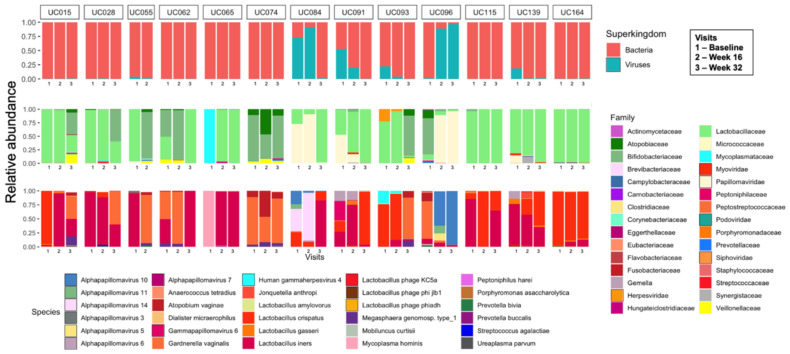
The vaginal microbial community composition of South African adolescents. Bar plot summarising the relative abundance of the 30 most abundant bacteria and viruses on kingdom, family, and species level identified by shotgun metagenomic profiling in vaginal samples collected at baseline and week 16 and from participants who did not have any sexually transmitted infections (STIs), including *C. trachomatis*, *N. gonorrhoea*, *T. vaginalis*, and *M. genitalium* throughout the study period, did not take any antibiotics or medication that might have influenced their microbiota, and did not practice vaginal insertional practices.

**Figure 2 viruses-13-02341-f002:**
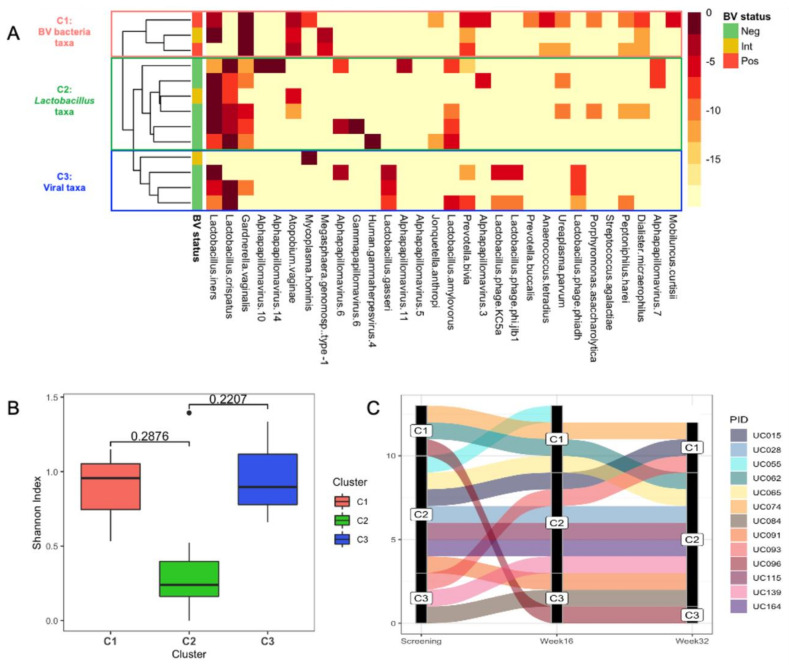
Vaginal microbial community clusters in South African adolescents. (**A**) Metagenomic data from baseline samples of eligible adolescents (n = 13) grouped into community clusters established using Manhattan clustering. Cluster 1 (C1) is dominated by a diverse range of BV-associated bacteria, C2 by *L. iners*, and C3 by a range of microbes including *Lactobacillus* and BV-associated bacterial spp. as well as viruses. BV status was assessed by Nugent Scoring. (**B**) Box-and-whisker plots depicting alpha diversity for each baseline sample (Shannon Index) by community cluster. (**C**) Alluvial plot showing the change in in community cluster from baseline to week 16 to week 32, coloured by participant ID (PID).

**Figure 3 viruses-13-02341-f003:**
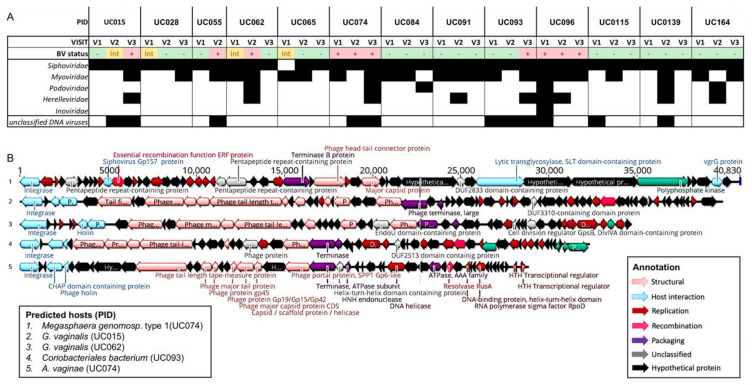
Prokaryote-infecting viruses in the vaginal metagenomes of South African adolescents. (**A**) Presence (black square) and absence (white square) of prokaryote-infecting viral families identified by shotgun metagenomic profiling of vaginal samples based on BLASTx analysis using the NCBI viral Refseq database. (**B**) Examples of circular/full-length Siphoviruses identified in adolescents with bacterial vaginosis (BV). Sequences were annotated using RAST and genes colour-coded based on function. Predicted hosts were determined and participant ID (PID) indicate in which participant they were present.

**Figure 4 viruses-13-02341-f004:**
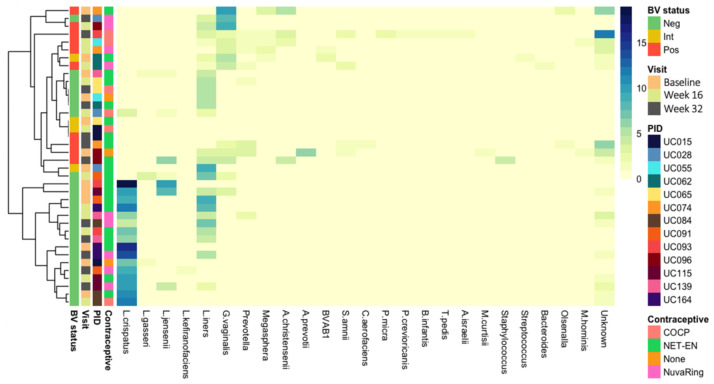
Presence of prophages in vaginal metagenomes of South African adolescents. The heatmap shows the number of “confident” and “likely” prophage-like elements identified by VirSorter in the metagenomes of the 13 South African adolescents included in this study. Host bacteria were identified by aligning the flanking regions of the predicted prophage-like element with the NCBI bacterial Refseq database using BLASTx. Samples are annotated by BV status (assessed by Nugent Scoring), visit (baseline, week 16, and week 32), participant ID (PID), and hormonal contraceptive use at time of sampling (combined oral contraceptive pills [COCP], the injectable NET-EN, none, or the combined contraceptive vaginal NuvaRing^®^).

**Figure 5 viruses-13-02341-f005:**
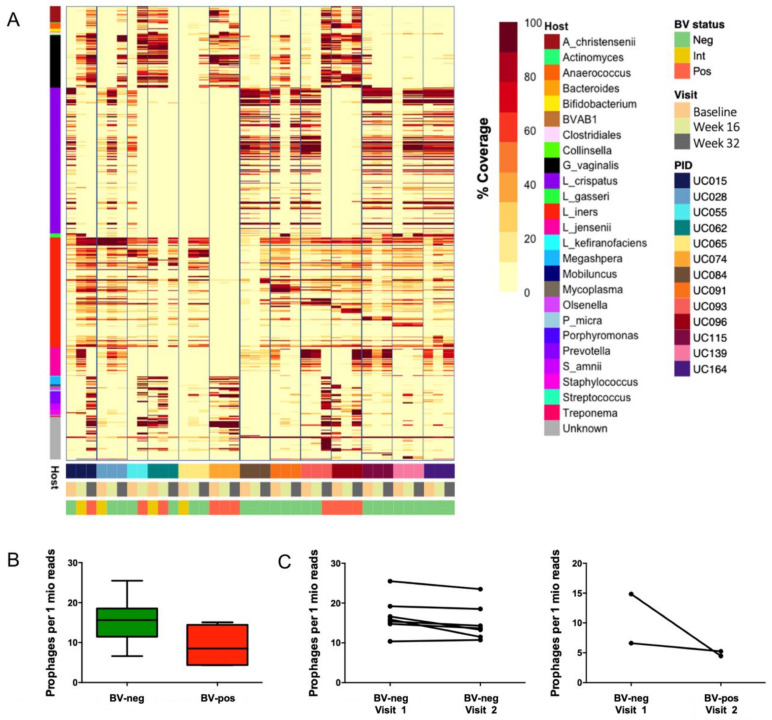
Persistence of prophages in vaginal metagenomes of South African adolescents and association with microbiota composition and stability. (**A**) Prophage-like elements identified by VirSorter were mapped against all reads in the dataset to evaluate the persistence of prophage-like elements over time. Samples are annotated by predicted bacterial host, BV status (assessed by Nugent Scoring), visit (baseline, week 16, and week 32), and participant ID (PID). (**B**) Box-and-whisker plots showing the number of prophages per million reads of participants who did not have Nugent-BV (BV-neg; n = 8) or those who had Nugent-BV (BV-pos; n = 4) at the final visit. (**C**) Prophage number per million reads of participants who remained Nugent-BV negative and those experiencing a change in Nugent-BV status from week 16 to week 32.

**Figure 6 viruses-13-02341-f006:**
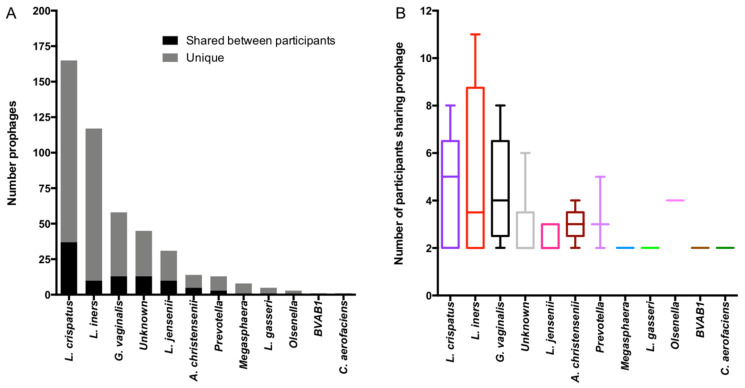
Sharing of putative prophages between participants. (**A**) Ninety-six putative prophages were identified in cervicovaginal secretions of more than one participant with ≥99% nucleotide identity, and the number of shared and unique (only present in a single participant) prophages are displayed by identified bacterial host. (**B**) The number of participants sharing prophages of a given bacterial host is shown using box-and-whisker plots.

**Figure 7 viruses-13-02341-f007:**
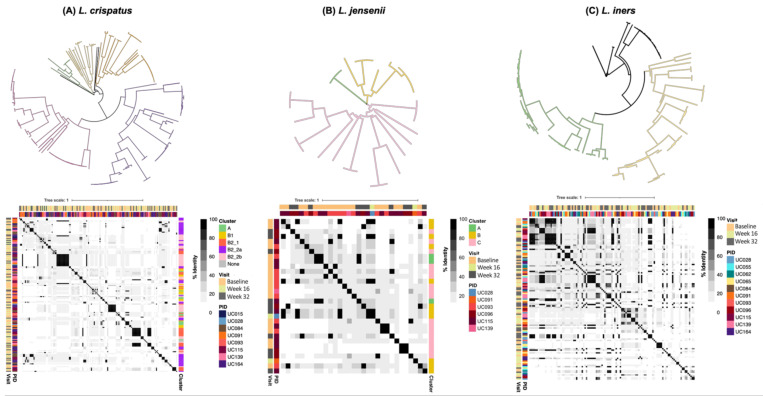
Diversity of *Lactobacillus* prophages identified in vaginal metagenomes of South African adolescents. To identify prophage clusters, multiple alignment of all sequences from (**A**) *Lactobacillus crispatus*, (**B**) *L. jensenii*, and (**C**) *L. iners* prophages was performed, a correlation matrix generated using the percentage alignment, and a tree generated using FastTree and visualized using iTOL. Each correlation matrix was annotated by identified prophage cluster, visit (baseline, week 16, and week 32), and participant ID (PID).

**Figure 8 viruses-13-02341-f008:**
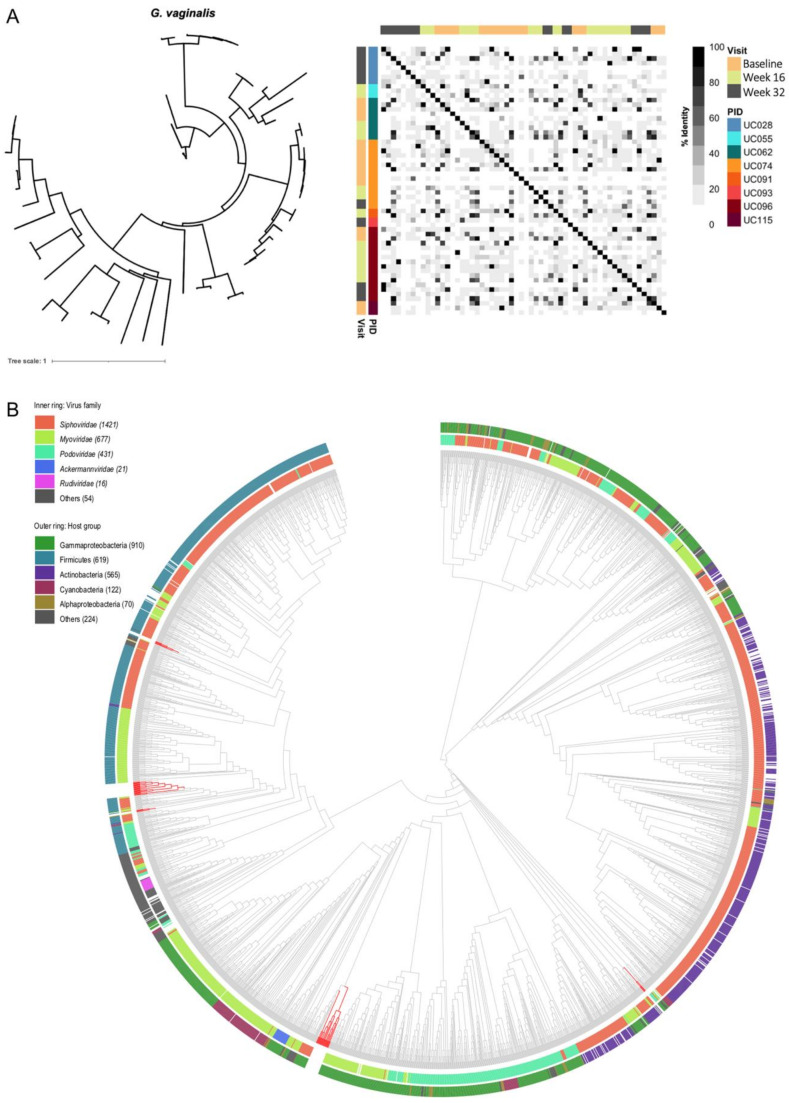
Diversity of *G. vaginalis* prophages identified in vaginal metagenomes of South African adolescents. (**A**) To identify prophage clusters, multiple alignment of all sequences from *G. vaginalis* prophages was performed, a correlation matrix generated using the percent alignment, and a tree generated using FastTree and visualized using iTOL. The correlation matrix was annotated by visit (baseline, week 16, and week 32), and participant ID (PID). (**B**) A proteomic tree including the identified putative *G. vaginalis* prophages was generated using VipTree. Red branches indicate the putative *G. vaginalis* prophages from this dataset, while grey branches indicate the viruses included in the reference database. The inner ring annotates the family of a given virus, and the outer ring identifies the host group.

**Figure 9 viruses-13-02341-f009:**
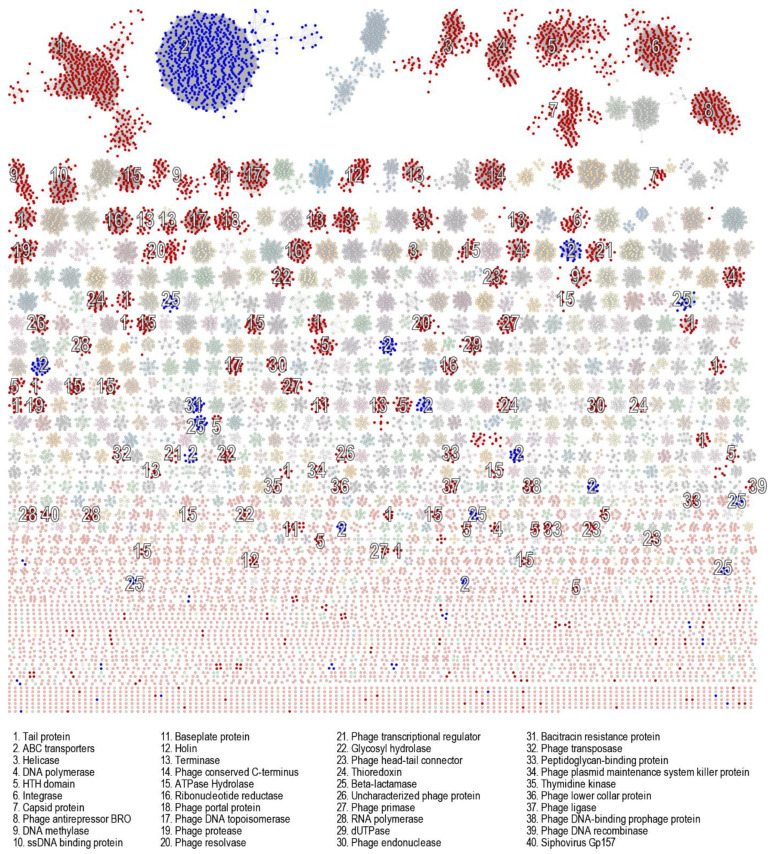
Sequence similarity network of proteins encoded by the identified prophages. Translated prophage protein sequences were used to generate a sequence similarity network (SSN) using EFI-EST. The network was visualized using Cytoscape and protein clusters of bacteriophage hallmark genes (red; including terminase, capsid, integrase, tail, portal and baseplate proteins, head–tail connector, holin, XRE-family HTH domain, DEAD-like helicase, DNA primase/helicase [DnaB], DNA polymerase A and B, DNA gyrase B, DNA topoisomerase IV, DNA ligase, DNA methylase, ribonucleotide reductase, thioredoxin, UvrD-like helicase, AAA family ATPase, holliday junction resolvase, HNH endonuclease, dUTPase, phage antirepressor BRO, peptidoglycan recognitions protein, ssDNA-binding protein, thymidine kinase) and potential antibiotic resistance genes (blue; including ABC transporters, beta-lactamase, and bacitracin resistance protein) highlighted.

**Table 1 viruses-13-02341-t001:** Cohort characteristics.

	Baseline n = 13	Week 16 n = 13	Week 32 n = 12
Age, median years (IQR)	16 (16, 17)	-	-
BMI, median (IQR)	23.6 (22.0, 25.1)	23.9 (22.5, 24.9)	24.3 (23.5, 26.3)
BV prevalence, n (%)			
BV+ (Nugent 7–10)	8 (61.5)	8 (61.5)	8 (66.7)
BV intermediate (Nugent 4–6)	3 (23.1)	1 (7.7)	0 (0.0)
BV− (Nugent 0–3)	2 (15.4)	4 (30.8)	4 (33.3)
Yeast prevalence (Gram stain), n (%)	2 (15.4)	0 (0.0)	2 (16.7)
HSV-2 serology prevalence, n (%)	2 (15.4)	2 (15.4)	2 (16.7)
CST distribution, n (%) ^a^			
CST-I	4 (30.8)	5 (38.5)	4 (33.3)
CST-III	5 (38.5)	5 (38.5)	5 (41.7)
CST-IV	3 (23.1)	3 (23.1)	3 (25.0)
Shannon Index, median (IQR) ^a^	0.71 (0.38, 1.34)	1.49 (0.39, 1.75)	1.12 (0.89, 1.50)
Days since the last menstrual period, median (IQR)	66 (23, 93)	19 (12, 54)	19 (10, 21)
Age menarche, median (IQR)	13 (13, 14)	-	-
Previously pregnant, n (%)	1 (7.7)	-	-
Hormonal contraception at time of sampling, n (%)			
None	1 (7.7)	0 (0.0)	0 (0.0)
Injectables (NET-EN/DMPA)	12 (92.3)	3 (23.1)	5 (41.7)
COCP	0 (0.0)	5 (38.5)	2 (16.7)
CCVR	0 (0.0)	5 (38.5)	5 (41.7)
Sexual risk behaviour			
Age of sexual debut, median [IQR]	15 [14, 16]	-	-
Any sexual partner(s) past year, n (%)	11 (91.7)	-	-
Multiple sexual partners past year, n (%)	1 (7.7)	-	-
New partner past year, n (%)	4 (33.3)	-	-
Condom use during last penile-vaginal intercourse, n (%)	9 (75.0)	7 (70.0)	9 (75.0)
Reported alcohol use, n (%)	2 (15.4)	2 (15.4)	4 (33.3)
Reported cannabis use, n (%)	1 (7.7)	0 (0)	0 (0)

^a^ Based on vaginal bacterial microbiota assessment by 16S rRNA gene sequencing [51]. CST-I = *L. crispatus*-dominated, low diversity; CST-III = *L. iners*-dominated, low diversity; CST-IV = diverse groups of anaerobic BV-associated bacteria, high diversity. BMI, body mass index; BV, bacterial vaginosis; CST, community state type; CCVR, combined contraceptive vaginal ring; COC, combined oral contraceptive pills; HSV, herpes simplex virus.

**Table 2 viruses-13-02341-t002:** Identified prokaryote-infecting virus families, genera, and species.

Family	Genus	Species (Number of Adolescents in Which Identified)
*Siphoviridae*	*Unclassified*	Stx2-converting phage 1717 (1)
		*Streptococcus* phage phiARI0468-2 (2)
		*Streptococcus* phage phiARI0462 (1)
		*Streptococcus* phage phiARI0031 (1)
		*Streptococcus* phage PH10 (3)
		*Streptococcus* phage MM1 (1)
		*Streptococcus* phage Dp-1 (2)
		*Streptococcus* phage 5093 (1)
		*Enterococcus* phage vB_EfaS_IME197 (1)
		*Lactococcus* phage WRP3 (1)
		*Lactococcus* phage Q54 (1)
		*Lactococcus* phage bIL311 (2)
		*Lactobacillus* phage PLE2 (1)
		*Lactobacillus* phage phiJB (1)
		*Cronobacter* phage ENT39118 (1)
		*Clostridium* phage phiCP39-O (2)
		*Clostridium* phage phiCD211 (2)
		*Brevibacillus* phage Sundance (1)
		*Bacillus* phage vB_BtS_BMBtp3 (1)
		*Bacillus* phage vB_BanS-Tsamsa (1)
	*Andromedavirus*	*Bacillus* virus Blastoid (1)
	*Ceetrepovirus*	*Corynebacterium* virus Zion (7)
	*Doucettevirus*	*Propionibacterium* phage E6 (2)
	*Magadivirus*	*Bacillus* phage Mgbh1 (1)
	*Moineauvirus*	*Streptococcus* virus Sfi19 (2)
		*Streptococcus* virus phiAbc2 (1)
	*Poushouvirus*	*Corynebacterium* phage Poushou (1)
	*Sextaecvirus*	*Staphylococcus* phage 6ec (1)
	*Spbetavirus*	*Bacillus* virus SPbeta (1)
*Myoviridae*	*Unclassified*	*Bacillus* virus G (4)
		*Enterobacteria* phage phi92 (1)
		Shigella phage SfIV (2)
		*Sphingomonas* phage PAU (3)
		*Streptococcus* phage EJ-1 (2)
	*Abouovirus*	*Brevibacillus* phage Abouo (1)
	*Firehammervirus*	*Campylobacter* virus CP21 (1)
		*Campylobacter* virus CPt10 (1)
	*Peduovirinae*	*Pseudomonas* phage phi3 (1)
		*Escherichia* phage pro147 (1)
	*Punavirus*	*Escherichia* virus P1 (1)
		*Salmonella* phage SJ46 (1)
	*Vequintavirinae*	*Klebsiella* phage vB_KpnM_KB57 (1)
*Podoviridae*	*Lederbergvirus*	*Salmonella* phage vB_SemP_Emek (1)
	*Picovirinae*	*Streptococcus* phage Cp1 (2)
		*Enterococcus* phage EF62phi (1)
	*Uetakevirus*	*Escherichia* phage TL-2011b
*Inoviridae*	*Unclassified*	*Propionibacterium* phage B5 (1)
*Herelleviridae*	*Unclassified*	*Lactobacillus* virus Lb338-1 (1)
		*Brochothrix* phage A9 (1)
	*Bastillevirinae*	*Bacillus* phage Deep Blue (3)
	*Brockvirinae*	*Enterococcus* phage EFDG1 (3)
	*Spounavirinae*	*Bacillus* virus SPO1 (1)
		*Bacillus* phage Shanette (2)

## Data Availability

The raw sequences of bacteriophage data presented in this study are openly available in NCBI under BioProject PRJNA767784, SRA reference numbers SRR16503456, SRR16503446, SRR16503439, SRR16503457, SRR16503452, SRR16503445, SRR16503443, SRR16503460, SRR16503451, SRR16503458, SRR16503459, SRR16503450, SRR16503427, SRR16503447, SRR16503449, SRR16503444, SRR16503437, SRR16503454, SRR16503455, SRR16503453, SRR16503442, SRR16503448, SRR16503441, SRR16503440, SRR16503435, SRR16503438, SRR16503433, SRR16503434, SRR16503430, SRR16503436, SRR16503429, SRR16503431, SRR16503432, SRR16503428, SRR16503426, SRR16503425, SRR16503424, SRR16503423.

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
