# Peer review of "Presence and Persistence of Putative Lytic and Temperate Bacteriophages in Vaginal Metagenomes from South African Adolescents"

_viruses, 2021, doi:10.3390/v13122341_

Round 1
Reviewer 1 Report
- "persistence" was used in this study. What is mean? Which characteristics of phage does the word indicate?
- in the abstract, "(pro)phages......may influence vaginal bacterial community composition" was described. Why not bacteria affect the phage composition?
- "130 nonpregnant HIV-seronegative females aged 15–19 years were enrolled". Why only Longitudinal samples from 13 participants were selected?
- the Fig 7, pictures of evolutionary branch analysis (up) is not distinct.
- Fig 9, Can authors further process it? It looks a little chaotic.
Author Response
We thank the reviewer for the thoughtful suggestions and are providing point-by-point answers below:
"persistence" was used in this study. What is mean? Which characteristics of phage does the word indicate?
In this study, persistence was defined as the presence of identical prophages (when limiting the required nucleotide identity to ≥ 99%) at two subsequent sampling time points, indicating that the bacterial host strains they are integrated in were stable colonisers of the genital tract. We have added a definition of persistence to the manuscript in lines 743-745.
in the abstract, "(pro)phages......may influence vaginal bacterial community composition" was described. Why not bacteria affect the phage composition?
Previous literature has primarily described that bacteriophages play crucial roles in shaping the composition and diversity of bacterial communities by infecting their prokaryotic hosts. We however acknowledge that this relationship may not only be unidirectional since the available bacterial hosts that are present within an environment may influence which bacteriophages are present. We have amended the abstract and added this to the discussion.
"130 nonpregnant HIV-seronegative females aged 15–19 years were enrolled". Why only Longitudinal samples from 13 participants were selected?
As stated in lines 144 – 152, we selected only samples from participants who did not take any antibiotics or medication that may have influenced their microbiota throughout the study period nor 40 days prior to enrolment, who did not practice any vaginal insertion practices (including washing with water and/or soap, insertion of traditional or other medicines, cloth, tampons and douching), had not been diagnosed with an STI during the study, and had attended all study visits. Unfortunately, only 13 of the 130 participants fit these criteria. We have now also added this explanation to the result section (lines 565-569).
the Fig 7, pictures of evolutionary branch analysis (up) is not distinct.
Thank you, we have now included higher resolution figures in the manuscript.
Fig 9, Can authors further process it? It looks a little chaotic.
We have processed this figure further to make its appearance less chaotic.
Reviewer 2 Report
Happel et al. present a descriptive study showing distribution of virulent bacteriophages, prophages and CRISPR loci in vaginal samples of South African women.
They did a comprehensive analysis of the phageome in vaginal environment, by metagenomics analyses of bacterial hosts, CRISPR analyses, prophages and phage-associated antibiotic resistance elements.
The paper is pointing out few novel findings, that are limited due to the very low number of enrolled patients, only 13, and to the very restricted age span of the patients (16 year old).
Regarding numbers, the authors carefully specify that no conclusive statement can be done, based on such limited sample size.
Regarding the age, see major comments.
Major comments
The paper should be more specific referring to the cohort, not as “women”, but adolescents or at least “young women”, and both title and text should be changed accordingly.
Data about smoking status of the patients is missing; according to recent literature, smoking habits and benzo[a]pyrene molecules were associated with induction of prophages in the vagina, and it should be mentioned in the first table.
Line 158-163: Can you check the following:
“All aligned read 158 data were subject to the following steps: (1) “duplicate removal” (i.e., the removal of reads 159 with duplicate start positions..............Remaining 162 reads were assembled using metaSPAdes”
could it be that this was included by mistake? From my understanding, you are correctly removing the human reads, but this part of methods refers to alignment of them, instead of removal. Can you explain?
Fig 4: I suggest to show a hierarchical clustering of this graph, if it serves the purpose of displaying a clusterization of the individuals based on their status of BV- or BV+. Same goes for 5A.
Line 367:
“These results further suggest that the presence of prophages within the 366 genomes of vaginal bacteria contributes to the stability of the vaginal microbiota.” This sentence should include at least a biological hypothesis that motivates it.
Did you check that patients using Nuvaring were clustering separately from patients receiving injections or oral contraceptives? due to the nature of the device, it could disrupt the microbiome.
Minor comments
Please check English and typos in the text
(e. g. Line 254: did not *taken* any antibiotics or medication that may have *influence*; Line 273 the change *in* in community cluster)
Please change here and in the text, bacterial names to italic.
E.g. line 342 “L. crispatus (n=165), L. iners (n=117), G. 342 vaginalis (n=58) and L. jensenii (n=31), L. gasseri (n=5), Prevotella spp. “
Line 337: “Identification of putative prophages in the metagenomes of South African young 337 women and associations with vaginal microbiota stability: We next evaluated ...”
the sentence before the semi column stands like a title, but is missing the italic formatting, or should be eliminated.
Author Response
We thank the reviewer for the thoughtful comments and suggestions and are providing a point-by-point response below.
Major comments
The paper should be more specific referring to the cohort, not as “women”, but adolescents or at least “young women”, and both title and text should be changed accordingly.
Thank you, we have amended this throughout the manuscript.
Data about smoking status of the patients is missing; according to recent literature, smoking habits and benzo[a]pyrene molecules were associated with induction of prophages in the vagina, and it should be mentioned in the first table.
Thank you for this great suggestion. Unfortunately, no data on cigarette smoking was collected as part of the parent study. We have added the prevalence of alcohol and cannabis use to Table 1 and comment on the lack of collection of smoking status in the discussion.
Line 158-163: Can you check the following:
“All aligned read 158 data were subject to the following steps: (1) “duplicate removal” (i.e., the removal of reads 159 with duplicate start positions..............Remaining 162 reads were assembled using metaSPAdes”
could it be that this was included by mistake? From my understanding, you are correctly removing the human reads, but this part of methods refers to alignment of them, instead of removal. Can you explain?
Thank you – your understanding is correct and we have rectified this in the method section.
Fig 4: I suggest to show a hierarchical clustering of this graph, if it serves the purpose of displaying a clusterization of the individuals based on their status of BV- or BV+. Same goes for 5A.
Thank you for this suggestion. We have clustered Figure 4 as suggested but prefer keeping Figure 5 ordered by visit since its purpose is to show the persistence of prophages over time within a participant.
Line 367:
“These results further suggest that the presence of prophages within the 366 genomes of vaginal bacteria contributes to the stability of the vaginal microbiota.” This sentence should include at least a biological hypothesis that motivates it.
We have amended this section and added our hypothesis. (Now line 756-759)
Did you check that patients using Nuvaring were clustering separately from patients receiving injections or oral contraceptives? due to the nature of the device, it could disrupt the microbiome.
Thank you, this is a great suggestion. We have previously described based on 16S rRNA gene sequencing that Nuvaring use indeed caused more pronounced shifts in the vaginal microbiota of adolescents compared to oral contraceptives or injectables (https://doi.org/10.1038/s41467-020-19382-9). We attempted similar analyses with the dataset of the sub-study, but probably due to the small sample size, were not able to see any separation of Nuvaring users from adolescents using injections or oral contraceptives. We have added this to Figure 4 and the discussion.
Minor comments
Please check English and typos in the text
(e. g. Line 254: did not *taken* any antibiotics or medication that may have *influence*; Line 273 the change *in* in community cluster)
Thank you, we have reviewed the manuscript and corrected typos and grammatical errors.
Please change here and in the text, bacterial names to italic.
E.g. line 342 “L. crispatus (n=165), L. iners (n=117), G. 342 vaginalis (n=58) and L. jensenii (n=31), L. gasseri (n=5), Prevotella spp. “
We have corrected this accordingly.
Line 337: “Identification of putative prophages in the metagenomes of South African young 337 women and associations with vaginal microbiota stability: We next evaluated ...”
the sentence before the semi column stands like a title, but is missing the italic formatting, or should be eliminated.
We have corrected this accordingly.